# LHC-like Proteins: The Guardians of Photosynthesis

**DOI:** 10.3390/ijms24032503

**Published:** 2023-01-28

**Authors:** Guy Levin, Gadi Schuster

**Affiliations:** 1Faculty of Biology, Technion, Haifa 32000, Israel; 2Grand Technion Energy Program, Technion, Haifa 32000, Israel

**Keywords:** photosynthesis, photoinhibition, light-harvesting, photoprotection, non-photochemical quenching

## Abstract

The emergence of chlorophyll-containing light-harvesting complexes (LHCs) was a crucial milestone in the evolution of photosynthetic eukaryotic organisms. Light-harvesting chlorophyll-binding proteins form complexes in proximity to the reaction centres of photosystems I and II and serve as an antenna, funnelling the harvested light energy towards the reaction centres, facilitating photochemical quenching, thereby optimizing photosynthesis. It is now generally accepted that the LHC proteins evolved from LHC-like proteins, a diverse family of proteins containing up to four transmembrane helices. Interestingly, LHC-like proteins do not participate in light harvesting to elevate photosynthesis activity under low light. Instead, they protect the photosystems by dissipating excess energy and taking part in non-photochemical quenching processes. Although there is evidence that LHC-like proteins are crucial factors of photoprotection, the roles of only a few of them, mainly the stress-related psbS and lhcSR, are well described. Here, we summarize the knowledge gained regarding the evolution and function of the various LHC-like proteins, with emphasis on those strongly related to photoprotection. We further suggest LHC-like proteins as candidates for improving photosynthesis in significant food crops and discuss future directions in their research.

## 1. In Addition to Functioning in Light Harvesting, the Light-Harvesting Proteins Are Important for Protection from Photoinhibition

Light-harvesting chlorophyll-binding complexes (LHCs) evolved in eukaryotic photosynthetic organisms to enhance the absorption cross-sections of photosystems I and II (PSI and PSII). LHCI and LHCII form megacomplexes with PSI and PSII (PSI-LHCI and PSII-LHCII) in thylakoid membranes and harvest light energy via the bound chlorophylls *a* and *b*. The harvested light energy is then directed towards the photosystem’s reaction centres (RCs) to drive charge separation and photosynthesis (Figure 1).

Owing to their significant enhancing effect on the ability of PSI and PSII to harvest sufficient light for efficient photosynthesis, LHCs have enabled eukaryotic photosynthetic organisms to evolve and thrive in various light-limited habitats. However, under intensive high-light (HL) intensities or when photosynthetic electron flow is inhibited, the efficient antenna property of LHCs is problematic and could increase photoinhibition. When the PSs are saturated with absorbed energy, the chlorophylls embedded in the LHC are excited with more energy than can actually be exploited by the photochemical reaction centres, resulting in the formation of chlorophyll triplets. These chlorophyll triplets react with O_2_ to form harmful reactive oxygen species (ROS) (Figure 1). In a similar fashion, ROS is produced in the RC of the PSII and PSI, possibly causing photoinhibition (PI), a condition in which photosynthesis is inhibited since the PS, primarily PSII, is damaged faster than the rate of being repaired (Figure 1 and Figure 2) [1,2,3]. During PI, photosynthesis declines up to a point where the organism may not survive. Photosynthetic organisms employ various mechanisms to minimize the effects of PI, starting from reduction in antenna (LHC) size to limit harvesting of excess light [4,5], through energy dissipation via non-photochemical quenching (NPQ) processes [6,7], and up to a rapid repair of the damaged photosynthetic apparatus [8,9,10] (Figure 1 and Figure 2). While LHCs function mainly as antennas that harvest light energy via chlorophylls and funnel it to the RCs (Figure 1 and Figure 5) [11,12,13,14], they also participate in the quenching of excess energy. An important and well described example of its role in energy quenching is the activation of the highly protective xanthophyll cycle during exposure to HL intensities. Violaxanthin, a carotenoid-bound to LHCII, is de-epoxidized by violaxanthin de-epoxidase (VDE) in a two-step manner first to antheraxanthin and then to zeaxanthin. Zeaxanthin then quenches the excess energy of nearby excited chlorophyll and prevents the production of harmful ROS, thereby protecting the PSII (Figure 1).

LHC proteins contain three transmembrane (TM) helices, of which two are homologous to HL-inducible protein (HLIP) (Figure 3a,b) and harbour a conserved amino acid sequence known as the LHC motif, as well conserved carotenoid locations and light-harvesting chlorophyll binding sites [15]. Intriguingly, it was discovered that LHC proteins evolved from proteins that do not participate in light harvesting (Figure 4), but instead serve to protect the PS RCs from excess light (Figure 1, Figure 2 and Figure 5) [16,17]. These LHC-like proteins share many structural similarities with the LHC proteins (Figure 3a,b), contain LHC motifs in their TM helices (Table 1), and are mostly able to bind pigments.

## 2. LHC-like Proteins Do Not Serve as Antenna but Function in Protection against Photoinhibition

The first known ancestor of LHC and LHC-like proteins was discovered in the HL-stressed cyanobacteria Synechococcus and was named HL-inducible protein (HLIP) [19]. It is now thought that HLIP was involved in a series of gene fusions, duplications, and deletions, which gave rise to all other LHC-like and LHC proteins, with the latter being the most recently formed group [17,20] (Figure 4). The LHC-like proteins are classified into four groups based on their structural characteristics and the number of TM helices they carry (Table 2). Group 1 includes proteins with one TM helix. They include HLIPs in cyanobacteria and one-helix proteins (OHP) or small chlorophyll a/b binding (CAB) proteins in eukaryotes. Group 2 includes proteins with two TM helices, known as stress-enhanced proteins (SEP) or LHC-like proteins (Lil). Group 3 includes proteins with three TM helices, which contain both early light-induced proteins (ELIP) and stress-related LHCs (lhcSR). Group 4 contains a protein with four TM helices named PSII subunit S (psbS) [20]. LHC-like proteins are generally similar to LHCs in their amino acid sequence and also share structural similarities (Figure 3) [16,20,21]. However, as mentioned above, they do not participate in light harvesting to drive photosynthesis. On the contrary, there is evidence that at least some of these proteins have critical roles in the quenching of excess energy in the NPQ reactions (Figure 1, Figure 2, and Figure 5 and Table 2). Among these, the most studied are psbS and the stress-related LHCs (lhcSR), which are present in land plants and algae, respectively, while mosses carry both of them [6,22,23] (Figure 5). LHC-like proteins are generally not detectable in non-stressed environmental conditions, and are induced by various stress factors, including cold temperatures, limited nutrients, desiccation, and intense HL [24,25,26]. Consequently, it was suggested that LHC-like proteins take part in adaptation to growth-stress conditions. The LHC-like proteins, which are induced by HL, with emphasis on those reported to function in photoprotective mechanisms, are the focus of this review.

While the photoprotective roles of psbS and lhcSR are well defined (Figure 5) [23], little is known about the photoprotective mechanisms of other LHC-like proteins. The known and suggested photoprotective roles of LHC-like proteins include the sensing of thylakoid lumen acidification, activation of the photoprotective xanthophyll cycle, chlorophyll and carotenoid synthesis regulation, repair of PSI and PSII, and more [28,29,30,31]. Together, LHC-like proteins act as the guardians of photosynthesis in a variety of stress conditions. As such, it is important to explore the molecular mechanisms underlying their photoprotection capabilities. Doing so will enhance our understanding of PI resistance, and may identify candidates for improving photosynthesis in agriculture crops. For example, transgenic expression of *Arabidopsis* psbS alongside the key enzymes of the xanthophyll cycle VDE and zeaxanthin epoxidase (ZEP) in tobacco lead to faster NPQ relaxation kinetics during a shift from HL to LL conditions, which, in turn, accelerates CO_2_ assimilation recovery in the shade. Importantly, both field and greenhouse experiments resulted in a 15% increase in crop productivity, as measured by dry mass, and a 33% increase in seed yield [32,33,34]. It is of note that PsbS could not be detected in one of the most light-tolerant algae on the planet, Chlorella ohadii, and lhcSR is entirely absent from its genome [5,35], raising questions as to how this alga so efficiently ensures PI protection and why it eliminated these PI protection players in its evolution.

Herein, we review the suggested general characteristics and photoprotective roles of LHC-like proteins. We explore the potential of improving photosynthesis via the genetic engineering of photoprotective LHC-like proteins in agriculturally important crops, discuss the importance of a gaining a better understanding of the photoprotective functions of LHC-like proteins, and suggest future research directions.

## 3. lhcSR and psbS Are Stress-Induced Proteins That Protect Photosynthesis against Photoinhibition

Similar to LHC proteins, lhcSR contains three TM helices (Figure 3). It is found mainly in various algae lineages and mosses and binds chlorophylls and carotenoids [36,37]. In the model green alga *Chlamydomonas reinhardtii,* three genes encode lhcSR: lhcSR 1, and lhcSR’s 3.1 and 3.2, which the later share the same amino acid sequence. Excessive light conditions trigger lhcSR expression and their accumulation within several hours [38]. Protonatable conserved Glu and Asp residues located at the luminal side of lhcSR sense thylakoid lumen acidification [39], which occurs as a result of proton intake during the elevated activity of the photosynthetic electron flow under HL intensities (Figure 1 and Figure 5). This acidification activates q_e_, the pH-dependent component of NPQ [40]. lhcSR binds chlorophyll *a*, lutein and violaxanthin, which under HL is converted to zeaxanthin (via antheraxanthin) through consecutive de-epoxidation events, a process known as “the xanthophyll cycle” (Figure 1). While lhcSR associates with the LHCs, zeaxanthin quenches the excited chlorophylls and thereby prevents ROS formation in both PSI and PSII [41] (Figure 1 and Figure 5).

In land plants, psbS is the pH sensor and not lhcSR. However, while lhcSR has three TM helices, psbS contains four (Figure 3) and while it does carry LHC motifs, structural analysis of *Spinacia oleracea* psbS revealed a structure that is too tightly packed to bind pigments such as chlorophylls [42]. Therefore, psbS may be responsible for proton sensing and the activation of q_e_ (the quenching of energy due to lowering the lumen pH) in higher plants, but it cannot be the quencher itself. Instead, it likely undergoes a conformational shift that facilitates its interaction with and activation of other LHC proteins for energy quenching, with recent studies indicating the minor antenna CP29 [43] (Figure 5). Interestingly, mosses contain both lhcSR and psbS, where both contribute to NPQ [44] and although initially psbS could not be detected in the model green algae *C. reinhardtii* under a variety of growth conditions [45], it was recently shown to be transiently induced under HL conditions. However, it was reported to accumulate to low levels, and to play a minor role in NPQ compared to lhcSR [46,47]. Surprisingly, psbS appears to be continuously expressed upon exposure to low light (LL) levels during a diurnal cycle [48,49]. Considering this evidence, psbS in *C. reinhardtii* was recently suggested to participate in the initial activation of lhcSR and NPQ, as well as in the reorganization of thylakoid membranes and LHC proteins [50]. Furthermore, the accumulation of lhcSR 1 and psbS in the thylakoid membrane of *C. reinhardtii* is induced by UV-B light using the nucleo-cytosolic UVR-8 photoreceptor. Accumulation of lhcSR 1 and psbS in UV-B light leads to a significantly better photoprotection under subsequent HL stress, which induces the accumulation of lhcSR 3 [51]. While most studies have focused on the protection of PSII via energy dissipation, under HL conditions the UV-B induced lhcSR 1 mediates energy transfer from LHCII to PSI instead of PSII, thus protecting it from excess energy [52].

## 4. One-Helix LHC-like Proteins (HLIPs/OHPs/Small-CABs) Protect PSII during Biogenesis and the Photoinhibition Repair Cycle

HLIPs were discovered in HL-stressed cyanobacteria [19] and are believed to be the earliest ancestor of all other LHC and LHC-like proteins [17,20]. HLIP homologs were later also found in eukaryotes [53] and were named OHPs or small chlorophyll *a/b* binding proteins (small-CABs). They possess a single trans-membrane helix, which harbours an LHC motif (Figure 3 and Table 1) and pigment-binding sites [54]. In both cyanobacteria and eukaryotic photosynthetic organisms, HLIPs/OHPs were shown to play central roles in the biogenesis and repair of PSII. In the cyanobacteria *Synechocystis* and *Synechocococcus*, there are four HLIP genes (*HliA-D* or *ScpB-E*) [55] which encode proteins that bind PSII and stabilize the trimeric structure of PSI, imperative for its activity under HL conditions [56,57,58]. HLIPs have been shown to retard PSII-associated chlorophyll degradation and have been suggested to temporarily bind the pigments of PSII during its repair, when its subunits are being substituted [56,59,60] (Figure 1 and Figure 2). In line with these reports, recent findings discovered HLIPs as part of chlorophyll-binding protein complexes that are important in the early assembly steps of PSII and chlorophyll biosynthesis [61,62]. A model of HLIP predicted a homo-dimeric structure and it was demonstrated that energy is dissipated via direct energy transfer between chlorophyll *a* and β-carotene [54]. The proteins may function by dissipating excess light energy that is absorbed by chlorophylls during the PSII repair cycle, when it cannot be utilized for photochemistry (Figure 1 and Figure 2).

In eukaryotic organisms, OHP1 and OHP2 replace HLIPs and are similarly induced by HL [53,63]. Recent studies suggest that they are crucial for PSII assembly and chlorophyll biosynthesis [30,64,65,66,67]. Mutations of either OHP1 or OHP2 in *Arabidopsis* resulted in less pigmentation, a disrupted thylakoid architecture and severe growth deficits, [66] while a deletion of OHP1 affected the function of core PSII proteins and resulted in lower accumulation of PSI protein [30]. Further studies indicated that OHP1 and OHP2 form a temporary protein complex with high chlorophyll florescence 244 (HCF244) and core proteins of PSII during PSII biogenesis [64,67] and that OHP function is fully dependent on its pigment-binding capacity [67]. These findings indicate that similarly to HLIPs, OHPs may temporarily associate with PSs and bind pigments while damaged core proteins are being replaced (Figure 1 and Figure 2).

## 5. Two-Helix LHC-like Proteins (SEPs/Lils) Function in Chlorophyll Biosynthesis and Protection against Photoinhibition

SEPs/Lils likely evolved when an early eukaryotic OHP-like protein acquired a second TM helix, which did not contain the LHC motif (Figure 3 and Figure 4). Later, different SEPs gave rise to LHCs, ELIPs, and psbS through gene duplications and, in the case of LHC and ELIP, an eventual loss of the fourth helix [17,20] (Figure 4). Four SEPs were identified in *Arabidopsis*: the poorly studied HL-induced Lil4 and Lil5 (formerly known as SEP1 and SEP2, respectively) and Lil3.1 and Lil3.2, which are present at constant levels regardless of the light regime [15,68]. Although its expression is independent of light, Lil3 forms a homo-multimeric chlorophyll-binding protein complex in dark-grown barley seedlings that are de-etiolated (transferred to light), suggesting light-dependent regulation of the complex formation [69]. Lil3 has been implicated in the stabilization of geranylgeranyl reductase (GGR) and has been found associated with protochlorophyllide oxidoreductase (POR), key enzymes in the production of crucial metabolites used for chlorophyll *a* biosynthesis [70,71]. Its participation in chlorophyll biosynthesis was strengthened by additional studies in barley and *Arabidopsis* [72,73]. Finally, Lil3 binds pigments and is capable of NPQ via direct energy transfer from chlorophyll *a* to a xanthophyll (lutein or zeaxanthin), suggesting that similar to HLIPs, it can also transfer pigments to their final destination while quenching damaging light energy [74]. In contrast to HLIPs/OHPs, Lil3 associates with LHCII subcomplexes rather that the PS core, which may be an indication of its distinct targets for photoprotection during PI [72] (Figure 2).

## 6. Three-Helix LHC-like Proteins (ELIPs) Play Important Roles in Protection against Photoinhibition

Like SEPs, ELIPs co-localize with the LHC and, similar to lhcSR and LHC, contain three TM helices and accumulate during HL stress [20,75,76,77]. ELIPs bind chlorophyll *a* and numerous molecules of lutein and have therefore been suggested to scavenge excited chlorophylls that are unbound to LHC during LHC protein turnover and thereby prevent ROS formation [78,79] (Figure 1 and Figure 2). Two ELIPs were identified in *Arabidopsis*: ELIP1 and ELIP2 [68]. An *Arabidopsis* mutant (chaos) that cannot rapidly accumulate ELIPs, suffered from bleaching and photooxidative damage during exposure to HL, due to ROS formation by excited chlorophylls [80]. The constitutive expression of ELIP genes prior to HL stress was sufficient to prevent these affects [80]. Moreover, the constitutive expression of ELIP2 resulted in a decrease in the levels of chlorophyll, and of glutamyl tRNA reductase and magnesium chelatase, which are key enzymes in chlorophyll biosynthesis [31]. Together, these observations suggest that ELIPs bind the chlorophyll of damaged LHCs while suppressing chlorophyll synthesis in order to avoid the accumulation of excited free chlorophylls during HL stress. This scenario was later challenged, as it was shown that a double elip1/elip2 mutant of *Arabidopsis* was just as prone to PI and photooxidative stress as WT and did not accumulate free pigments [29]. It was noted that zeaxanthin was decreased in the elip1/elip2 mutant, perhaps indicating that ELIP regulates the protective xanthophyll cycle. In an additional study, the enhanced protection of PSII from PI was noted in an *Arabidopsis* mutant lacking ELIP2 but transgenically expressing the ELIP1/2 of the desiccation-tolerant moss *Syntrichia caninervis* [81].

Ten ELIP homologs were identified in *Chlamydomonas*, some of which are induced by HL stress [82,83]. ELIP3 is upregulated by a combination of HL and cold stress, resulting in improved survival of *C. reinhardtii* under photooxidative stress and higher PSII activity [26]. In accordance, an ELIP3 knockdown mutant showed significantly less PSII activity and did not survive under HL and cold stress conditions [26]. Moreover, the addition of a ROS quencher reduced ELIP3 expression, suggesting that ROS trigger ELIP3 expression [26]. In the green alga *Dunaliella salina*, an ELIP homolog is co-regulated with carotenoid biosynthesis and was therefore named carotenoid biosynthesis-related protein (CBR) [84]. CBR associates with LHCII, binds zeaxanthin [85,86], and was suggested to be involved in the organization and/or stabilization of PI-protective protein complexes with high lutein and zeaxanthin content [87]. Interestingly, CBR is the most upregulated protein in HL-adapted *Chlorella ohadii* cells, one of the most light-tolerant photosynthetic organisms [5,27]. Analysis of CBR accumulation in *C. ohadii* cells grown under HL found co-localization of CBR and LHCII, and ranked it to be the most abundant LHC-like protein with a light-dependent expression pattern [27]. It is important to note that *C. ohadii* does not contain genes encoding lhcSR [35] and psbS was found to be expressed below the proteomic detection levels under HL conditions [5,27]. These results implied that CBR plays a key role in the photoprotection of photosynthesis in an extremely light-tolerant organism, which lacks lhcSR and expresses limited or no quantities of psbS. Intriguingly, *C. ohadii* CBR has been found to be homologous to the abovementioned ELIP3 of *Chlamydomonas*, which was shown to be induced by ROS formation and crucial for its survival under HL and cold stress [26,27]. Additionally, amino acid sequence analysis and structural modelling of CBR implied that a conserved glutamate is located in the same position as one of the proton-sensing glutamates of psbS, suggesting, in the absence of psbs and lhcSR, a possible role in sensing low pH in the lumen during HL (Figure 5).

## 7. Soluble Pigment Binding Proteins also Contribute to Photoprotection

As described, LHC and LHC-like proteins share distinct structural similarities, are located in the thylakoid membrane, and contribute to photoprotection. These proteins, with the exception of psbS, bind both carotenoids and chlorophylls. In addition, soluble pigment-binding proteins also offer alternate routes for energy dissipation via NPQ and thus promote photoprotection. The structures of these proteins differ significantly from the membranal LHC and LHC-like proteins and they do not share the LHC and LHC-like ancestor protein. In cyanobacteria, one such protein is the orange carotenoid protein (OCP) [88]. OCP forms a homodimer complex where each monomer binds a single carotenoid [89]. Upon activation by sensing blue-green light, OCP undergoes a conformational change and interact with the cyanobacterial photosynthetic antenna (phycobilisome). Following this, via its bound carotenoid, it presents a route for the thermal dissipation of excess energy and acts as a singlet oxygen quencher [90,91,92,93].

In plants, an example is the water-soluble chlorophyll-binding protein (WSCP) [94]. WSCPs form a homo tetramer and, contrary to the cyanobacterial OPC, bind one chlorophyll per each monomer, but no carotenoids [95]. The exact function of WSCPs remain a mystery; however, evidence suggests a role in photoprotection, despite the lack of carotenoids. Chlorophyll bound to WSCP is much less sensitive to intense light compared to unbound chlorophylls due to a formation of a physical barrier by the phytyl moieties, shielding the oxidizable sites of the chlorophyll, and thus preventing damage by ROS to the chlorophylls themselves [95,96,97]. Moreover, the overexpression of WSCP in transgenic plants leads to a reduction in ROS generation, highlighting its role as a protective protein under stress [98].

## 8. LHC-like Proteins Function as Enhancers of Photosynthesis and Survivability during Abiotic Stress and Their Overexpression May Increase Crop Yield

Considering the growth of the Earth’s population and the anticipated food shortage, new approaches to increase food production should be explored and developed. Improving photosynthesis as a means of enhancing crop growth and yields has been a long-standing aim [33,34,99,100,101]. Other techniques implemented to achieve this important goal include the selection of light-tolerant strains and improving the efficiency of CO_2_ assimilation and light-to-biomass conversion, as well as the manipulation of metabolic pathways by genetic engineering. Alongside their crucial roles in the protection of photosynthesis from photoinhibition, studies have suggested the potential of LHC-like proteins as photosynthesis enhancers. The overexpression of psbS in field-grown tobacco led to a decrease in stomatal opening in response to light, which resulted in a significant 25% reduction in water loss per assimilated CO_2_ molecule [102]. As described above, the transgenic expression of *Arabidopsis* psbS alongside VDE and ZEP in tobacco accelerated NPQ relaxation kinetics during a shift from HL to LL conditions, which, in turn, shortened time to CO_2_ assimilation recovery in the shade. Importantly, both field and greenhouse experiments resulted in a 15% rise in crop productivity, as measured by dry mass [32]. Moreover, using the same strategy in the important crop soybean grown in fields, leads to an increase of up to 33% in seed yield [34]. In rice, another highly important crop, psbS overexpression resulted in a higher grain yield under fluctuating light conditions [103]. These studies provide proof of the ability to increase biomass and yield in important crops via the overexpression of the LHC-like protein psbS, exclusively or alongside other proteins. While psbS is the most studied LHC-like protein, other members of the family were also shown to increase biomass and to provide protection under abiotic stress. In *C. reinhardtii* mutants lacking all lhcSRs, expression of lhcSR under the control of a heat shock protein promoter allowed for the fine-tuning of NPQ, resulting in higher photosynthetic efficiency and biomass accumulation [104]. Transgenic tobacco overexpressing ELIP showed higher survivability after freezing, a higher net photosynthesis rate after chilling, a higher plant fresh weight after osmotic stress, and higher maximal photochemical efficiency after HL treatment [105]. As mentioned earlier, the overexpression of ELIP3 in *Chlamydomonas* improved the survival of the cells under HL and cold stress. In contrast, a knockdown mutant showed low photosynthetic efficiency and rapidly died under these conditions [26]. Similarly, the overexpression of *Rhododendron* ELIP in *Arabidopsis* resulted in a much stronger tolerance to freezing and higher PSII function after 3 days of recovery from freezing [106]. A recent study in the red alga *Neopyropia yezoensis* overexpressing OHPs failed to detect any obvious enhancement of photoprotection [107]. In contrast, *Arabidopsis* OHP knockout mutants exhibited higher susceptibility to HL [108], were unable to grow on soil [30], and demonstrated compromised growth [66]. Taken together, these studies highlight the potential of using psbS and other LHC-like proteins to protect plants and improve photosynthesis under various abiotic stresses.

## 9. Concluding Remarks and Future Perspectives

The LHC-like proteins form a large group of mostly HL-induced TM proteins that are capable of binding pigments and performing NPQ via energy transfer from chlorophylls to carotenoids. An exception to this rule is psbS, which is unable to bind pigments due to its tightly packed structure, containing four TM helices [42] (Figure 3a). Instead, psbS is protonated upon acidification of the thylakoid lumen and undergoes a conformational shift before interacting with PSII and activating NPQ in the LHCII minor antenna CP29 [43] (Figure 5). Similarly, under acidic conditions, lhcSR is protonated and associates with LHCII proteins [39], but unlike psbS, it binds carotenoids that offer a route for the thermal energy dissipation of excess energy in the LHCs of both PSs [41] (Figure 1 and Figure 5). Similarly, ELIPs and SEPs associate with LHCII subunits and bind pigments. Their suggested role involves the temporary binding of chlorophylls that were detached from damaged LHC subunits during their turnover under HL stress, while simultaneously regulating chlorophyll synthesis [72,74,78] (Figure 1 and Figure 2). In *C. ohadii*, one of the most light-tolerant photosynthetic organisms on the planet [35], an ELIP homolog named CBR accumulates massively under HL and co-localizes with LHCII [5,27]. The absence of lhcSR from the genome of *C. ohadii* and the failed detection of psbS in both LL- and HL-grown cells [5,27] raise questions about the significance and function of CBR in this particular organism. It was suggested that CBR is activated in a similar manner to psbS and lhcSR, as they all contain a conserved protonable Glu on their luminal side [27] (Figure 5). Unlike the abovementioned proteins, HLIPs in cyanobacteria and OHPs in eukaryotes seem to associate with the PS core proteins, rather than with the LHC, where they are also suggested to act as a temporary reservoir of detached pigments from damaged core proteins during their turnover in HL stress [56,60,64,67] (Figure 1 and Figure 2). The functions of the LHC-like proteins discussed in this review are summarized in Table 2.

Although the exact molecular mechanisms through which LHC-like proteins act to protect the PSs from PI are yet to be fully uncovered, it is clear that they play key roles in photoprotection, which can be exploited to increase the yield and survivability of agriculturally important crops. Recent experiments have shown this potential by overexpressing psbS, resulting in increased crop productivity and grain yield while reducing water loss due to stomatal opening in plants exposed to fluctuating light [32,34,102,103]. These promising results highlight the biotechnological potential and importance of understanding and utilizing LHC-like proteins to improve photoprotection capacities. Notably, psbS is by far the most studied LHC-like protein. However, as described above, other LHC-like proteins have shown to increase biomass and protect photosynthesis and should definitely be further tested for their ability to also increase crop productivity. In the future, psbS and the other LHC-like proteins can be introduced to important crops via genetic engineering, both to increase their yield and expand their environmental niche by making them more resilient to light and other stresses.

## Figures and Tables

**Figure 1 ijms-24-02503-f001:**
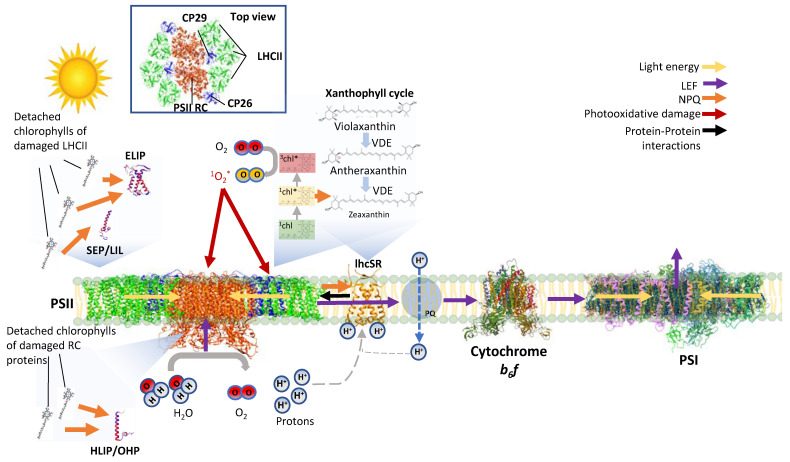
Photosynthesis during HL and suggested photoprotective roles of LHC-like proteins. Light is harvested in the LHC and directed to PSII RC, where it excites a specialized chlorophyll-pair (P680) and initiates charge separation and linear electron flow (LEF). Electrons are transferred from PSII-RC via QA to QB that then join the plastoquinone (PQ) pool. From PQ, the electrons move to cytochrome b6f, PSI, and ferredoxin NADPH reductase (FNR) (not shown). FNR reduces NADP+ to NADPH, which is further utilized for carbon-fixation reaction via the Calvin–Benson–Bassham cycle. During HL, PSI and PSII are saturated with energy. The continuous photosynthesis facilitates a proton gradient between the lumen of the thylakoid and the stroma of the chloroplast due to the splitting of water and import of protons via the PQ pool and cytochrome b6f. This lumen acidification leads to protonation of protonable residues in psbS (mostly plants and mosses) and lhcSR (algae and mosses). In response, the pigment-less psbS interacts and activates NPQ in the LHCII subunit CP29. lhcSR binds lutein and chlorophyll and can act as a sinkhole for excess energy in the LHC. Thus, it can receive and scavenge excess energy from LHCII. When the light energy exceeds the amount that could be used for photochemistry, excited chlorophylls in the LHCs and RCs may dissipate the excess energy as heat via interactions with other chlorophylls or carotenoids. During HL, the xanthophyll cycle is activated: violaxanthin is de-epoxidated by VDE to zeaxanthin (via antheraxanthin), which in turn acts to relax excited chlorophylls and prevent ROS formation. Remaining excess energy excites chlorophyll to a triplet state where it reacts with O_2_ to form ROS. The ROS then interacts with and damages the photosynthetic reaction centres and LHC subunits, leading to photoinhibition. It is suggested that SEPs/Lils and ELIPs serve as a temporary reservoir for the chlorophylls of the damaged proteins during their turnover (see also Figure 2).

**Figure 2 ijms-24-02503-f002:**
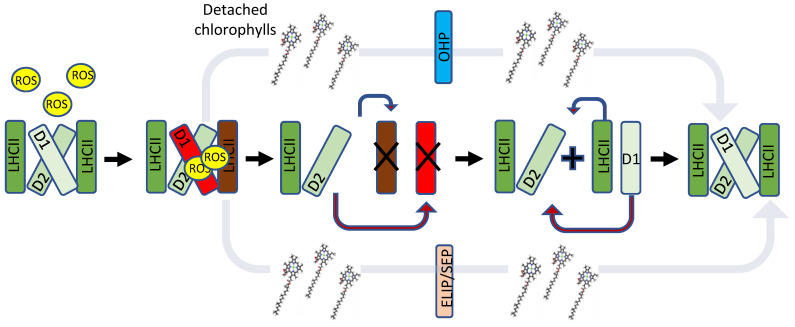
Suggested roles of HLIP/OHP, ELIPs and SEP/Lil in the PSII repair cycle. During HL, generated ROS damages mainly the PSII reaction centre protein D1, followed by D2, and then the LHCII subunits. Chlorophylls from the damaged reaction centres and LHC are transferred to OHP1 or ELIP and SEP/Lil, respectively, while the damaged proteins are degraded. During this time, the energy of excited chlorophylls is quenched via carotenoids. Once newly synthesized subunits are incorporated and PSII reassembled, chlorophylls are transferred back to the reaction centre to harvest light for photosynthesis. It is important to note that ELIPs, SEPs and OHPs are most certainly involved in regulation of other mechanisms, including chlorophyll and carotenoid biosynthesis.

**Figure 3 ijms-24-02503-f003:**
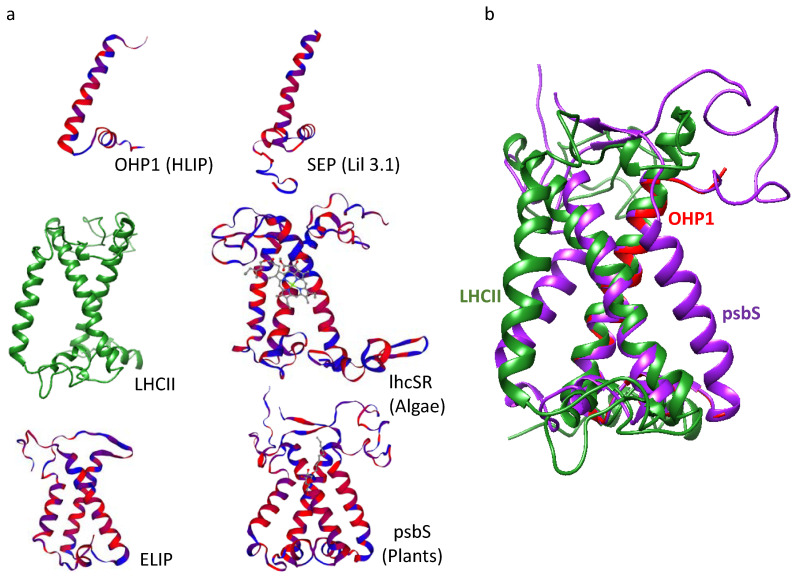
LHCII and LHC-like proteins share vast structural similarities. (**a**) The known structure of LHCII (taken from pdb id: 6KAF) is shown in green. The structure models of LHC-like proteins as predicted by Swiss model. (**b**) OHP1 and psbS predicted models superimposed on the known LHCII structure. Note the overlapping of the two conserved TM helices in OHP, LHCI and psbS. All models (including OHP and SEP) were predicted with *Arabidopsis thaliana* sequences of the relevant proteins except lhcSR. The lhcSR model was predicted with *C. reinhardtii* sequence. The predicted structures are coloured by hydrophobicity: red indicates hydrophobic regions and blue indicates hydrophilic regions.

**Figure 4 ijms-24-02503-f004:**
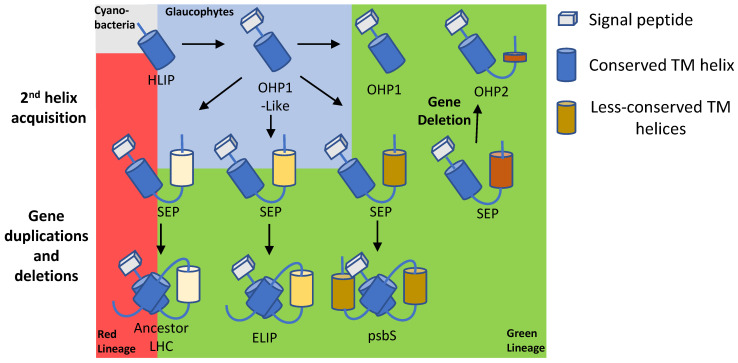
Suggested evolution of LHC and LHC-like proteins. HLIP was first introduced to eukaryotes via primary endosymbiosis and acquired a signal peptide when its gene transferred to the nuclear genome. OHP1-like protein gave rise to a pool of different SEPs. OHP2 is likely a consequence of a gene-deletion event of SEP while a small portion remains and acts as a membrane anchor. Different SEPs evolved into LHCs, ELIPs and psbS.

**Figure 5 ijms-24-02503-f005:**
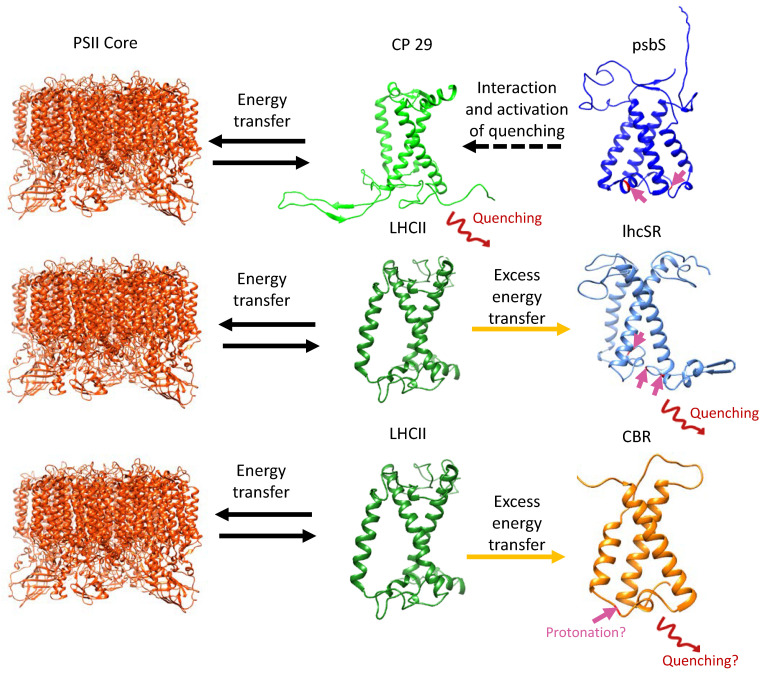
Proposed quenching mechanisms for psbS, lhcSR, and *C. ohadii* CBR. HL conditions promote thylakoid lumen acidification. psbS (mainly in plants) is protonated at conserved Glu residues (indicated by pink arrows) and undergoes conformational shift before interacting with and activating energy quenching in the LHCII subunit CP29 (top). lhcSR is protonated at conserved Glu and Asp residues and also undergoes conformational shift and interacts with LHCII subunits. Excess energy can be transferred to lhcSR, where it is quenched via carotenoids (middle). The mechanism of CBR is unknown, yet evidence points to a conserved Glu residue with the same location of one of the proton-sensing Glutamates of psbS, suggesting a possibility in which CBR acts as a substitute for psbS and lhcSR in the highly light-tolerant *C. ohadii* (Levin et al., 2022) [27]. Figure adapted and modified from Pinnola., 2019 [23] LHCII and PSII core structures were taken from pdb id: 6KAF.

**Table 1 ijms-24-02503-t001:** Conserved LHC motif in selected photoprotective LHC-like proteins. The symbols * in the consensus sequence indicate less conserved residues. Adapted and modified from Rochaix and Bassi, 2019 [18].

Organism	Protein	LHC-Motif Sequence	Location
*C. reinhardtii*	psbS2	ELFVGRLAMVGFSAS	71–85
		ELFVGRAAQLGFAFS	159–173
*C. reinhardtii*	lhcSR1	EITHGRVAMLAALGF	81–95
		ELNNGRLAMIAIAAF	191–205
*C. reinhardtii*	lhcSR3	EITHGRVAMLAALGF	87–101
		ELNNGRLAMIAIAAF	197–211
*C. reinhardtii*	ELIP (1)	EINNGRIAMVSVVTA	67–81
		EKINGRAAMMGLTSL	346–360
*C. reinhardtii*	ELIP (2)	EIVNGRLAMLGFVSA	103–117
		ELLNGRAAMIGFAAM	171–185
*C. reinhardtii*	Lil3 (SEP)	EKLNGRAAMMGYVLA	162–176
*A. thaliana*	Lil3 (SEP)	ELLNGRAAMIGFFMA	174–188
*A. thaliana*	OHP1 (HLIP)	EIWNSRACMIGLIGT	69–83
*A. thaliana*	OHP2 (HLIP)	EISNGRWAMFGFAVG	130–144
*Synechocystis*	HliA (HLIP)	EKLNGRLAMIGFVAL	36–40
	Consensus	E**NGR*AM*G	

**Table 2 ijms-24-02503-t002:** Summary of LHC-like proteins’ suggested pigments and functions.

TM Helices	Protein	Organism	Pigments	Suggested Function
1	HLIP (Prokaryota)OHP (Eukaryota)	Cyanobacteria, Algae, Plants	Chlorophyll a, β-carotene	Temporarily binds pigments during PSII repair cycle.
2	SEP/Lil	Algae, Plants	Chlorophyll a, Lutein/Zeaxanthin	Temporarily binds pigments during LHC turnover, chlorophyll biosynthesis.
3	ELIP	Algae, Plants	Chlorophyll a, Lutein, Zeaxanthin	Temporarily binds pigments during LHC turnover, suppression of chlorophyll biosynthesis
3	lhcSR	Algae and Mosses	Chlorophyll a, Lutein	Thylakoid lumen acidification sensing, destination for excess energy transfer from LHC.
4	psbS	Mostly Plants and Mosses	-	Thylakoid lumen acidification sensing, activation of NPQ in LHCII subunit CP29.

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
