# Peer review of "LHC-like Proteins: The Guardians of Photosynthesis"

_ijms, 2023, doi:10.3390/ijms24032503_

Round 1

Reviewer 1 Report

The authors summarize the evolution and function of the various LHC like proteins, and focus on the function in photoprotection. The MS was summarized comprehensively and logically.

Some concerns need to be addressed:

1.       The function of LHCSR1 upon UV radiation in Chlamydomonas should be mentioned and the corresponding ref should be cited.

2.       The resolution of Figure1 needs to be improved definitely, and the labelling of ‘D2’ in Figure 2 needs to be rearranged. The same problem occurs in Figure 4.

Author Response

Thankyou very much for your comments. 

  1. The function of LHCSR1 upon UV in Chlamydomonas is now described and cited.
  2. The figures were corrected. The resoulutions were humpered during the conversion to the PDF version. The figures look fine in the word version and the production team will use this version. 

Reviewer 2 Report

“LHC-like proteins: the guardians of photosynthesis” completed by Levin and Schuster has summarized, from view of the structure and function of proteins, and the evolution of LHC-like proteins. They have proposed that the LHC-like proteins are the guardians of photosynthesis from their photoprotection roles, and suggested that the LHC-like proteins can serve as candidates for improving crop products.

The review is interesting and significant in research of the structure and function of LHC protein family and searching for ways to improve photosynthesis and enhance the crop production. The manuscript is well written and scientifically sound. However, the author should still consider following two points:

1.      There is another chlorophyll binding protein, e.g. water soluble chlorophyll binding protein, which also plays important roles in photoprotection. Is it possible to include this protein?

2.      The authors suggested that the LHC-like protein can be very important in enhancing the crop production. More evidence or information should be provided for this argument.

Therefore, I suggest the manuscript to be published after minor revision.

Author Response

We thank the reviwer for the important and thoughtful comments.

  1. The soluble pigmant binding proteins are now described.
  2. Evidences of using the LHC-like protein to increase crops yield are now better described